# 'You are too much in this modern world, that's why you are like this': Understanding perceptions of mental health among Somali women in London

**Caitlin Gonzalez, Elizabeth Humberstone🟢, Chris Willott🟢***

Department of Population Health Sciences, King's College London, London, United Kingdom

* chris.willott@kcl.ac.uk

## Abstract

This is a qualitative study exploring the perceptions of mental health among Somali women living in London. Participants, over the age of 18, female and identifying as Somali, were recruited from a community centre in West London. Seven participants were recruited and semi-structured interviews were conducted to better understand perceptions of mental health, care-seeking, treatment and different understandings of these issues within the Somali community in London. Interview transcripts were imported into NVIVO version 14 to be coded, and description-focussed coding and thematic analysis were used to interpret key themes. Concerns around judgement, shame and stigma are key issues affecting attitudes towards mental health in this group. Other important issues affecting conceptualisations and attitudes towards mental health were intergenerational differences, isolation from the community, stigma and secrecy. Participants also reported the importance of protective factors, particularly faith and family in their lives. This research adds to existing literature in exploring perceptions of mental health in Somali communities in the UK and such research is helpful in identifying cultural barriers to recognition and treatment of mental health within this community.

## Introduction

### Background and context

According to the 2021 United Kingdom (UK) census, there are approximately 176,000 people living in England and Wales who identify as Somali [1]. Greater Somalia has experienced civil unrest since the start of the war in 1991, with millions of Somalis fleeing to parts of Africa, Europe and North America. Individuals fleeing conflict are significantly more likely to experience mental health problems due to trauma, settlement stress and social isolation. Higher prevalence of anxiety and depression is reported in Somali people in comparison to the general population in Western countries [2,3]. Furthermore, research shows that mental health

**Data Availability Statement:** All data underlying the findings reported in this article have been shared through data repository OSF. The identifier for the dataset is DOI 10.17605/OSF.IO/29CEF.

**Funding:** The author(s) received no specific funding for this work.

**Competing interests:** The authors have declared that no competing interests exist.

conditions are heavily stigmatised in Somali communities in the West, including in Norway [2], the United States [4] and the UK [5].

Somali people trace their origins to the Horn of Africa, and the main Somali population in Africa stretches across four countries (Somalia, Ethiopia, Kenya and Djibouti) and one *de facto* independent state (Somaliland). The region has been the site of numerous conflicts since the early 1980s, initially motivated by resistance to the military junta in the Somali Democratic Republic led by Mohammed Siad Barre [6]. Since then, numerous international, national and subnational conflicts involving state and non-state actors have affected the lives of millions of people. Conflict in the region continues to this day, alongside significant climate change-induced flooding [7]. Conflict and environmental change have caused mass emigration from the region, with displaced Somali people seeking refuge in other countries; whilst most have claimed asylum in neighbouring African countries, there are also large numbers of Somali refugees in Europe [8,9] (Fig 1).

## Somali diaspora in the UK and their relationship to mental health

Although official counts refer only to Somali people from the Federal Republic of Somalia rather than including ethnic Somalis from other African countries, evidence suggests that the UK is home to the highest number of Somali people in Europe [5].

Research conducted in the UK has found a high proportion of Somali people having unmet needs in key areas, such as housing, food, physical health, psychological distress and educational attainment [10]. Through discrimination and adversity, these factors, alongside experiences of conflict, make Somali people more susceptible to mental health symptoms [11,12].

Understanding Somali people's perceptions of mental health is crucial to creating culturally sensitive mental healthcare services, an area where people's needs are currently unmet [11,13,14]. Mental health service use has been noted to be low among Somali people living in London, despite a high proportion of the community reporting mental health problems [10].

Mental health service use for Somali people in Western countries may be limited due to a combination of stigma, lack of awareness at the community and family level and cultural preferences in seeking help in religion and community [15]. Thus, research into Somali attitudes toward mental health that explore the barriers to seeking formal mental health services and seek opinions on helpful routes for treating mental health could help support and develop services to become more culturally appropriate. Amongst the Somali diaspora, cultural, religious and moral values contribute to a strong sense of identity [16]. Religious practices, peer opinion and family support may all impact perceptions of mental health and it is generally expected that individuals would seek support from the community before engaging with formal mental health services [16,17].

There is an absence of words for psychological diagnoses such as depression or stress in the Somali language. This lack of vocabulary, alongside a low level of knowledge of Western psychological concepts, creates difficulty in having conversations concerning mental health [16]. Furthermore, research has suggested that restriction in vocabulary may contribute to a practice where individuals describe somatic symptoms such as headaches, tiredness and heart palpitations to express their mental distress [18]. Understanding such practices could therefore help support clinicians in primary care services in assessing and managing Somali patients appropriately.

Due to unfamiliarity with Western psychiatric labels, mental health is frequently viewed as binary; an individual is either 'sane' or 'insane' with no grey area in between, and those with mental health conditions are frequently labelled 'mad' or 'crazy' [5,19]. This dichotomy could contribute to stigma surrounding mental health, a concern that is noted within the community [17]. People with mental health conditions are also frequently seen as a risk to the community, further exacerbating stigmatising attitudes [11], and as a result, people with mental health

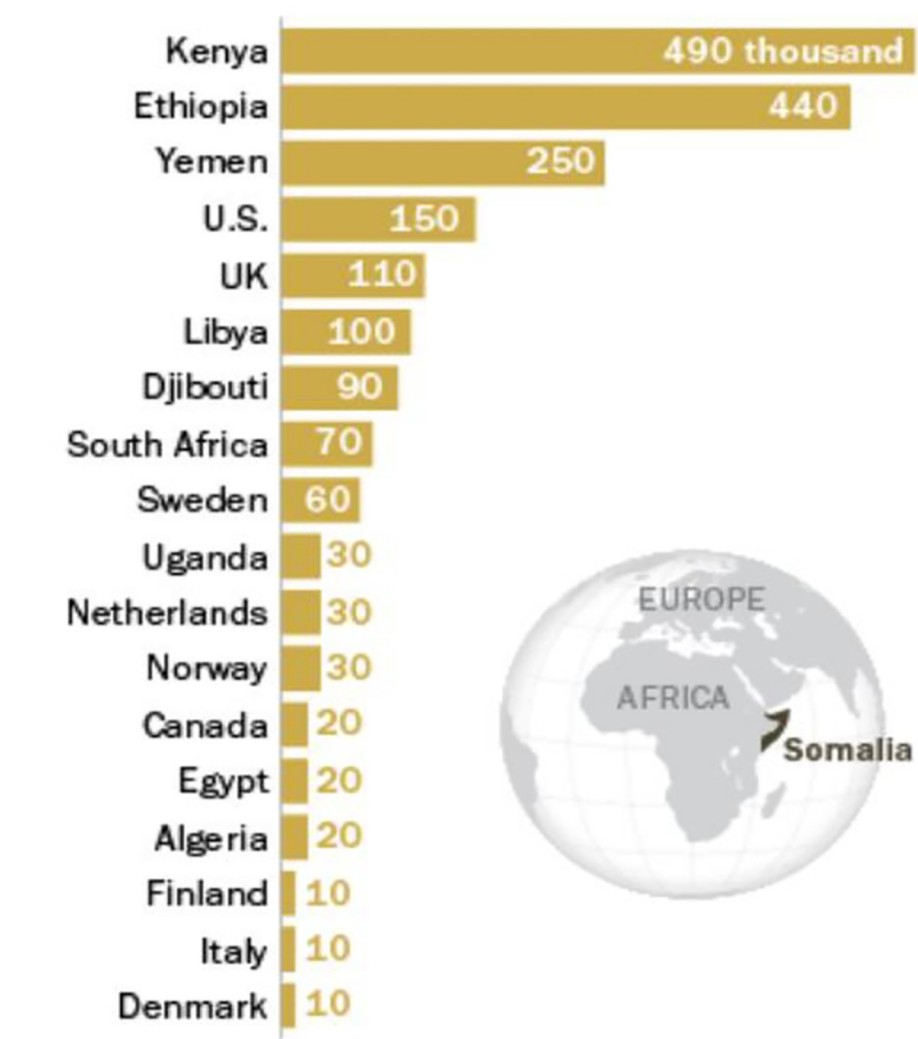

**Fig 1. Location of the Somali diaspora.** Pew Research Center (https://www.pewresearch.org/short-reads/2016/06/01/5-facts-about-the-global-somali-diaspora/). Accessed 16 February 2024.

conditions may be ostracised for speaking about their mental health condition or treatment, further reinforcing a cycle of shame, stigma and isolation [17,19].

## Materials and methods

### Study setting and rationale

In this article, we analysed the perceptions of mental health among Somali women living in London. The study participants all attend Ilays Community Centre in Feltham or are

connected to the centre in some way. Ilays is a non-profit organisation that helps black and minority ethnic (BAME) people arriving in the UK and was set up in 2004 by the East African community in West London [20].

Most current research on Somali communities in the Western world focuses on the United States. This research has identified a gap in the literature by further exploring perceptions of mental health among Somali women living in London. It will aid the understanding of the changes necessary to improve mental health services for this population group, and will contribute to knowledge of the perceptions, beliefs and practices prevalent among Somali women, while recognising culturally specific perspectives that affect help-seeking behaviours.

## Research paradigm and question

The main goal of the article is to analyse the meanings people attach to mental health. A qualitative approach was chosen as this provided an unconstrained, flexible and open environment for participants to respond to questions considering the nature and subject of the research [21].

This research uses an interpretivist approach where phenomena are observed through subjective accounts and opinions based on the individual's perception of the world in different contexts, allowing the researcher to create meaning and synthesise perceptions of mental health [22,23]. The ontological approach assumes multiple truths and realities and suggests that perceptions of mental health are socially constructed and subject to constant change [22]. This research is therefore intended as a sample of the views of Somali women living in London at a particular point in time.

## Data collection and sampling

Purposive sampling was used to recruit participants who are over 18, female and of Somali ethnicity living in London. This sampling method was used as it aims to select participants who are most likely to provide useful and appropriate information in interview [24]. Ilays was chosen as the basis for recruitment as this is a hub for Somali community activities in West London. CG visited the community centre several times to recruit participants using a poster. This was not successful so staff at the Centre put CG in touch with potential participants who were followed up with an information sheet.

This study recruited seven participants. We had originally intended to interview men and women but following extensive attempts at recruiting men, none came forward so we continued with the project and interviewed women only. A key reason why we argue that seven interviews is sufficient for this study is that we are not seeking generalisability nor theoretical saturation. We are aiming to provide rich descriptions and analyses of the views of female members of London's Somali community on mental health and are not seeking to argue that this is what female London Somalis, as a group, believe. We are not seeking theoretical saturation because the study is not based on grounded theory and we are not seeking to develop theory on the basis of empirical evidence (see for instance Braun and Clarke [25]). It is also the case that our sample is relatively homogenous so a relatively small sample size will have a smaller impact on the quality and trustworthiness of the evidence and conclusions.

In this article we draw on the principle of data adequacy [26]. Data adequacy suggests that what is important is not the number of participants, but 'the quality and sufficiency of information as it provides close access to the richness of the subject matter'[27] (p.12). Levitt et al [27] conclude that there are two criteria upon which methodological integrity can be assured, which are the basis for the approach we have taken:

(a) fidelity to the subject matter, which is the process by which researchers develop and maintain allegiance to the phenomenon under study as it is conceived within their tradition of inquiry, and (b) utility in achieving research goals, which is the process by which researchers select procedures to generate insightful findings that usefully answer their research questions (p.2).

All participants were under the age of 60, and all but one had migrated to London from Somalia, either at a young age or as parents with their children. Semi-structured interviews with these participants were conducted. Six interviews were conducted over the phone, a technique which offers flexibility and ease for participants, given that most participants were mothers and lived in different areas in London [28]. One face-to-face interview took place at the Ilays community centre. Data was collected in June and July 2023. All interviews were carried out by CG.

## Data analysis

Collected data was imported into NVIVO version 14 to be coded. Line-by-line description-focused coding was used because a descriptive approach provided insight into participants' experiences and perceptions [29] cited in [28]. Thematic analysis was used to interpret the data by identifying themes and patterns based on common perceptions of mental health among Somali women [30].

## Ethics

Ethical approval was granted by the King's College London Research Ethics Committee and study recruitment ran from 05/06/2023 until 14/07/2023. An information sheet was provided to participants explaining the purpose of the study and written, informed consent was obtained. Three interviews were audio recorded. Four participants did not consent to being audio recorded so their interviews were recorded through note taking, including clarifying what participants had said by repeating their words back to them. Pseudonyms were assigned to each participant to protect confidentiality.

## Results

This research aims to understand the perspectives on mental health of a group of Somali women living in London. The open interview technique meant that a variety of questions were asked of participants, and different themes and opinions could be explored. There were some themes that became evident within the interviews, including the personal importance of mental health to the participants, the impact of shame and community judgement on perceptions of mental health and different conceptualisations of mental health in the community based on age and the place where someone grew up. The following section is based around these three core themes.

### Personal importance of mental health

When asked about the importance of mental health, all participants expressed the significance of mental wellbeing in their lives, and reported opinions that robust mental health was as important as their physical health needs.

> It is very important because if you don't take care of it, you have nothing left.

> When I compare my physical health to my mental health, I can't function or do anything [when I'm mentally unwell] (aged around 40).

One participant had lost a family member to mental illness, whilst others described the stories of friends or family who had suffered with such conditions. Despite valuing mental health, three out of the seven participants were unable to list formal mental health conditions when asked, with two participants describing a need for better education in mental health, either personally or within the community. For four participants, mental illness was discussed in relation to difficulty coping with life events, rather than any organic cause, with one participant saying, "when people choose that they can't handle [stress], it goes to their mental [health]" (aged 25). Several participants differentiated between mental health conditions that they considered 'severe' and associated with madness, and those which they considered less severe, giving examples such as depression.

> [I] do acknowledge and understand the word mental health but do they mean depression, anxiety, low mood [. . .] you could have a low mood and that could be your mental health (aged around 40).

> Yeah, there is no [such thing] as mental health, you know what I mean? Unless you are fully like a crazy person (aged 26).

The use of the word 'crazy' and discussions of 'madness' was identified in the interviews. Some participants felt that this was a label the community would attach to them if they spoke up about their mental health:

> You could be having like depression, or you can have like stress, or you don't even have to have like full mental health yet and they will still label you as the crazy one (aged 26).

Although this shows the concerns about stigmatisation from the community, it also demonstrates perception of a distinction between 'full' mental illness and conditions like depression. Prior research has found that Somali people tend to label people as 'sane' or 'insane' rather than perceiving mental health conditions as a spectrum [5]. This can stigmatise both high-functioning individuals with mental health problems, because their condition could therefore could be seen as unimportant, and those with more acute mental health problems, whose condition is labelled 'madness'. Six participants used the words 'crazy' or 'madness' to discuss mental health conditions. This may result from a lack of Somali translation for some mental health conditions, but also could relate to this distinction between 'sane' and 'insane'.

### Stigma and community judgement

One participant implied that the binary discourse around sanity and insanity contributed to the stigma surrounding mental health by linking public stigmatisation to concerns about individuals being able to fulfil their roles in society:

> They'll be like, she takes that [medication]. Why do you need that? Like she's gonna go crazy. How do you know she's not gonna kill your children when she gives birth? (aged 26).

Stigma was a theme which predominated in the interviews, with all participants reporting a stigma surrounding discussions on mental health in the community. Six out of the seven participants used the word 'shame' in their responses, and all spoke at length about the opinions of community members on one's mental health. Some participants described the effect of stigmatisation in isolating them from the community:

> They tend to ostracise a person that has [mental illness], sometimes they tend to make fun of you, and a lot of them they might say they are not practicing enough . . . prayers.

> You want to ask people for help and advice and first thing is you go to the community where you share the same values and norms [. . .] But then it doesn't really work out for the person because they are shamed (aged 28).

The first response here notes the ways that community judgement on mental illness may involve religion, with concerns that speaking out about mental illness may cause judgement on the person's religious practices and that mental illness may simply be a result of not praying enough. Religion can be a helpful coping mechanism for those with mental illnesses, but views like this could also further isolate individuals with mental illness from religious communities. The second response expresses the desire to seek help within the Somali community, demonstrating the importance of shared values to the individual, but remarking on the way that this could isolate an individual and undermine that desire for community connection.

Participants linked the idea of stigmatization to an intentional secrecy surrounding mental health conditions:

> The family [. . .] might tell you to hide it, don't tell any other person because as a community if someone gets to know it, they will just talk bad about . . . our family (aged late 20s).

This fear of judgement also affects the family's reputation within the community. This may offer further reasons for individuals to feel concerned about discussing mental health with others and reinforce the taboo around mental health. Indeed, the theme of family was spoken about by all participants, with a range of responses noted. For four participants, family was considered important for supporting mental health recovery, with these participants reporting that they would approach their families if they were struggling. For the other three participants, however, family was not viewed as a support, with one participant saying "family can be worse" and another reporting that disclosing mental health conditions to family could cause a person to fall out of favour. Given that research suggests that many Somali people will seek help from families and communities before mental health services [17], we should consider the impact that stigma may have on individual families and how it may affect their ability to care for a family member with a mental health condition.

It was also suggested that the stigma surrounding mental health could limit someone's opportunities, for example in work or marriage.

> If you're not married, they will be like 'oh, who wants to marry you, like you are crazy'.

> If you want to go to work, they will be like 'who wants to hire you, you are crazy' [. . .] so people try to keep that to themselves (woman aged 26 years old).

Reports like this demonstrate the effects of such stigma in limiting an individual's ability to interact within their community, citing limitations to their roles in families or employment. With such a negative potential impact, this participant suggests that stigma can impact the course of a person's life and contribute to the secrecy surrounding mental health conditions within the Somali community.

## Community diversity: Geography and age

In some interviews, comparisons were made to the treatment of mental health in Somalia, where some reported that people could be "chained to walls" or "hidden" in the family home.

Three participants highlighted secrecy around mental health by discussing the practice of sending those with mental health conditions back to Somalia for treatment. In their responses they described families telling the community that they had gone on holiday, gone to learn about their culture and religion or gone away for work or study in order to hide their mental health issues.

> If someone gets help in the [UK] and people are shaming them in the way of getting married or going to work or to do anything, of course you are going to choose to go to another country to get the help, and then come back and hide it (aged 26).

Participants suggested that there were differences between generations in their understanding and treatment of mental health conditions. Six participants reported instances of Somali elders denying the existence of mental health issues such as depression. Some participants explained this in relation to older generations being raised in Somalia whilst younger generations were more likely to be raised in the UK. It was argued by three participants that many older Somalis rejected Western interpretations of mental illness, particularly the medical model, in favour of a religious or spiritual conceptualisation of mental illness.

> Still the older generation, they don't believe stress [. . .] they don't believe mental illness, like they all, all they say is like go back to your God (aged 35).

The interview data highlighted this generational difference, with participants attributing this difference to either older generations not wanting to know about mental health or being limited by the ways they were brought up. Indeed, one participant suggested that elders attributed mental illness to 'westernised' perspectives and lifestyles:

> You are too much in this modern world, that's why you are like this (aged late 20s).

> Did you get that from the European countries? Did you get that from the UK? Don't bring that please because we don't believe [it] (aged 26).

Here, the medical model of mental illness is presented as culturally inappropriate by older Somalis, which could contribute to stigmatisation of mental illness by creating a dichotomy between 'local' cultural tradition and 'foreign' Western medical model. Interview participants did not perceive these two perspectives as mutually exclusive, but complementary, with three participants speaking about the importance of both medical treatment, such as medication and therapy, and cultural traditions, such as reading the Qur'an, praying and speaking to members of the community.

> Go to religion and the doctor and tell them–don't hide anything (aged around 50).

Indeed, religion played a significant part in all interviews; all participants were Muslim and considered Islamic perspectives on mental health. For most participants, this was connected to a religious duty to look after their mental wellbeing, with one participant saying, "Islam tells us to look after ourselves and tells us to get help if needed" (aged 28). Other participants described the importance of prayer in supporting their mental wellbeing, with religion a huge source of comfort, and one participant suggested that religious scholars could help to break the taboo and stigma surrounding mental illness by initiating discussions about mental health in the community. Spiritual concepts, such as Jinn or "black magic" were also mentioned in the interviews as being related to mental health. In contrast to the comfort of religious teaching,

these concepts appeared to contribute towards the stigma of mental illness and were used to dismiss the validity of mental health as a concept. Furthermore, one participant spoke briefly about the concept of mental illness being seen a form of moral punishment, saying that others might ask:

> Why is their family like this? Maybe they did something wrong, so they deserve it (aged around 50).

One participant described her experience as a mother to a child with a mental health problem, and reported particular concern about younger generations, suggesting that they were more susceptible to mental health problems due to bullying and educational learning needs. On the other hand, many of the participants spoke of the increased education concerning mental health among younger people, citing social media as both an educational service and a means of normalising mental illness within the Somali community. Furthermore, participants spoke positively about the provision of mental healthcare in the UK, the importance of free services such as therapy and the high number of Somali people working within healthcare. The participants thus spoke positively about the medical model of UK mental healthcare, and the importance of further understanding and education on the subject.

Results from these interviews identified a few key themes, with the most important being the personal importance of mental health in people's lives, the existence of mental health stigma within the Somali community in London and the importance of diversity within the community, particularly based on the interrelated issues of age and the place where someone was brought up, with older people brought up in Africa being much more likely to hold stigmatising views on mental health. By contrast, the people we interviewed, most of whom were of a younger generation, had far more open-minded and adaptable views on mental health.

## Discussion

Stigma is a key element of Somali conceptualisations of mental health, and a significant barrier to individuals seeking formal psychological help [12,31]. In their study of Somali women's access to mental health services in Australia, Said et al. conceptualised stigma as either public or internalised, suggesting that public shame around mental illness leads to self-stigmatisation and secrecy concerning mental illness [31]. This was also seen in our research, where participants used the word 'shame' to describe community responses to mental illness, and the theme of secrecy–not revealing mental health struggles to others–was explored in six out of the seven interviews. Said et al's research further shows that community shame in mental health can impact the reputation of families and individuals by suggesting a relationship between weak faith and mental illness [31]. This type of attribution within a religious community might also contribute to a sense of stigma or shame surrounding mental illness.

In our research, participants described the social impacts of stigma in limiting opportunities for employment and marriage. Michlig et al. similarly observed discussions in Somali focus groups which described people hiding their mental health problems to avoid perceived negative social consequences such as the threat of divorce [11]. Michlig's research suggests that these perceived social consequences may lead to self-management of mental health conditions rather than accessing formal psychological help, whilst other research has also concluded that such attitudes can hinder help-seeking behaviours [18,31,32]. Moreover, other barriers to medicalised mental healthcare may come from a scepticism with seeking help outside of the community due to concerns that this might be considered shameful or culturally inappropriate [19,33]. However, participants in our research did not report problems with the UK's medical

model of mental health support, and all participants spoke positively of therapy, hospitals and medications, mentioning that these could help support mental wellbeing. This contrasts with Said et al's work in Australia, where there was significant distrust of 'western' mental health services [31]. However, like in our work, the group that predominantly experienced mistrust were older Somalis. This suggests that in both contexts, younger Somalis are becoming better integrated into and accepting of 'western' models of mental health and care.

Alongside community preferences for community-based support, stigma could contribute to the practice of sending people with mental health conditions to Somalia to receive treatment. Research in Finland also found that the difference between medicalised models of mental illness in the Finnish healthcare system and Somali views of mental illness contributed to Somali people seeking mental healthcare abroad [34]. In our interviews, participants mentioned this practice, with some discussing the way that this might contribute to the secrecy surrounding mental illness. It has been observed that there is a cultural practice of sending individuals back to Somalia (sometimes referred to as *dhaqan ceelis*) if their behaviour is felt to be inappropriate, and concerns have been raised that this includes individuals with mental health conditions [35].

Differences have been observed in the Western understanding of mental health and the way the topic is understood 'back home' in Somalia, demonstrated by the denial of the existence of stress and depression in Somalia [16,31,36]. Despite recent improvements in the mental health training for Somali doctors, it is accepted that there is a lack of mental healthcare infrastructure in Somalia to support people with mental health conditions [37]. Thus, there is limited exposure to medical treatments, with traditional and religious healers acting as the main providers of mental healthcare [38]. Given this, Somali people who migrated later in life may bring support for practices learnt in Somalia with them to the UK. Indeed, research suggests that Somali people frequently keep traditional practices and health belief systems even after settling in a new country [2], which may explain the intergenerational differences in the understanding of mental health identified in our research.

These differing views may also contribute to the stigma surrounding mental health, with older generations conceptualising it as a product of Western culture [15]. Furthermore, problems of vocabulary may contribute to this generational divide, with an absence of words for mental health conditions in the Somali language, and reports of difficulty sharing such concepts even when interpreters are used [33]. Research on the elderly Somali population and their views of mental health are scarce, and more research would help to provide detail on this issue.

Many participants were able to convey the importance of religion to their understanding of mental health. There are established positive associations between mental health and religion, and prior research reports Somali people turning to faith, prayer and the Qur'an to help support their mental wellbeing [39]. In Somali regions of Africa, ilaajs–religious mental health facilities–are widespread and frequently the first choice of people seeking mental health services [40,41]. However, some participants did describe concerns around community responses to mental illness being seen as a distance from God, or moral weakness, a concern which has also been reported in prior research [16]. Although only one participant in this study brought up the concept of Jinn or possession, some researchers have reported concerns that spiritual conceptions of mental illness in Somali communities can further reinforce stigma [42].

Stigma is an important barrier to mental health treatment that requires examination to allow for better community support. Research shows that family and community support can be protective against mental illnesses such as depression [43,44]. Specifically, research on East Africans in the United States found social support to be a key component to promoting psychosocial wellbeing, suggesting the detrimental effect of stigma and social isolation [45]. Social

support has been posited as a means of 'buffering' stressful experiences, such as those, like trauma or discrimination, which migrants to the UK are more likely to have experienced [44,46]. In our interviews, participants discussed the effects of stigma on isolating them from the community, but most participants expressed a desire to have discussions about mental health within their families and communities, indicating the importance of dismantling stigma to create stronger community support.

## Conclusion

This research has identified some key themes surrounding perceptions of mental health among Somali people living in London. A key theme identified and reported by all participants was that of the shame and stigma surrounding mental health. On identifying this theme, further subthemes were observed, such as the impact of intergenerational differences, isolation from the community and the difficulties of integrating Somali and UK culture. Stigma around mental health was reported as both public and internalised shame and was consistent with prior research on Somali perceptions of mental health. Furthermore, participants discussed the impact of stigma in affecting their opportunities in employment and marriage and discussed the differences between the treatment of mental health in the UK and Somalia. Participants reported the importance of education on mental health within the community and suggested that religious and community leaders could help to dispel the stigma surrounding mental health. Religion was also discussed as a protective factor in mental wellbeing and promoted as a way of supporting those with mental health conditions. The significance of religion to Somali conceptions of mental healthcare is consistent with previous research and indicates the need for culturally appropriate mental health services, to help serve Somali communities in the UK. However, our work also indicates a significantly more positive attitude of younger Somalis towards 'western' mental health services than those experienced by older people within the Somali community.

Previous literature on the Somali community in the UK and their perceptions of mental health is somewhat limited, with much scope for further research. Opportunities for further research identified by this study include further studies on mental health stigma within Somali communities, research into intergenerational differences of opinion surrounding mental health, research identifying the mental health challenges for Somali adolescents and studies into the perception of different mental health treatments for Somali people. This is particularly important given the low service use in these communities and could offer further information on how best to organise mental health services to support and serve this community.

## Author Contributions

**Conceptualization:** Caitlin Gonzalez, Chris Willott.

**Investigation:** Caitlin Gonzalez.

**Methodology:** Caitlin Gonzalez, Chris Willott.

**Supervision:** Chris Willott.

**Writing – original draft:** Caitlin Gonzalez, Elizabeth Humberstone.

**Writing – review & editing:** Elizabeth Humberstone, Chris Willott.

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
