## [Decision Letter · Decision Letter 0]

6 May 2024

PONE-D-24-07335‘You are too much in this modern world, that’s why you are like this’: Understanding perceptions of mental health among Somali people in London.PLOS ONE

Dear Dr. Willott,

Thank you for submitting your manuscript to PLOS ONE. After careful consideration, we feel that it has merit but does not fully meet PLOS ONE’s publication criteria as it currently stands. Therefore, we invite you to submit a revised version of the manuscript that addresses the points raised during the review process.

Both reviewers point out the need for a critical review of the methods and findings in the manuscript. A thorough justification of the methods must be provided through a critical and philosophical lens in order to merit scientific publication. In particular, some narratives must be provided on the recruitment strategy, the data analysis process/rationalisation, as well as how this manuscript links to current discourse on mental health perceptions in minority groups. Reviewer 2 also suggests some revision to the title and both reviewers highlight inconsistencies in several areas which require your attention.

We look forward to receiving your revised manuscript.

Kind regards,

Lily Kpobi, Ph.D.

Academic Editor

PLOS ONE

4. We note that Figure 1 in your submission contain copyrighted images. All PLOS content is published under the Creative Commons Attribution License (CC BY 4.0), which means that the manuscript, images, and Supporting Information files will be freely available online, and any third party is permitted to access, download, copy, distribute, and use these materials in any way, even commercially, with proper attribution. For more information, see our copyright guidelines: http://journals.plos.org/plosone/s/licenses-and-copyright.

Reviewers' comments:

Reviewer's Responses to Questions

**Comments to the Author**

1. Is the manuscript technically sound, and do the data support the conclusions?

Reviewer #1: Partly

Reviewer #2: Partly

2. Has the statistical analysis been performed appropriately and rigorously? 

Reviewer #1: N/A

Reviewer #2: N/A

3. Have the authors made all data underlying the findings in their manuscript fully available?

Reviewer #1: Yes

Reviewer #2: No

4. Is the manuscript presented in an intelligible fashion and written in standard English?

Reviewer #1: Yes

Reviewer #2: No

5. Review Comments to the Author

Reviewer #1: Abstract:

It needs to be revised to be a standard one.

Method:

1. Why all women were selected for the research needs clarification;

2. What are the mental health conditions of these selected women?

3. Why is the sample only 7?

4. I found from the result section that none of the women had/had mental health conditions, as they mostly referred to their relatives or friends. In such a context, interviews with 7 women must be justified as it is difficult to establish an argument based on a very small number of interviews. This can be a deciding factor if it is to be accepted or rejected by the editor. Thus, we need very clear and justified responses in the manuscript.

Result:

1. Quotes: Please use a citation or (add some basic information about the respondent of the quote, i.e., 60 years old woman;

2. Organising the results into two or three subsections based on the key themes would make the section clear and easy for the reader.

Discussion:

1. Participants in the research did not discuss traditional healers. However, the author discusses this issue, which is unrelated to the result. Instead, enriching the discussion around the spiritual issues and interpretations of mental illness would make the last part of the discussion interesting.

Reviewer #2: I have pointed some of the key issues to attend to:

1. Under data collection and sampling section of the manuscript, it is stated clearly that “this study recruited seven participants, who were all women”. However, the title is presented as perceptions of mental health among Somali people in London. Given the fact that this is a qualitative study, where voice is fundamental, it is advisable that the title adequately reflects the voices captured in the study. I recommended ‘Somali women in London’ instead of ‘Somali people in London’.

2. The sample of seven women should also be clearly included in the abstract for clarity.

3. In the abstract, it is stated that ‘description-focused thematic analysis’ was used to interpret key themes. Under data analysis section of the methodology, it is stated that description-focused coping was done, and then thematic analysis was used to identify themes and patterns. This presents difficulty and inconsistency regarding what constitutes thematic analysis and description-focused thematic analysis.

4. The data analysis as presented in the methodology is scanty and does not present enough information to convince readers of the rigor of the analysis.

5. Respondents and participants are interchangeably used in this manuscript. There is the need to stick to interpretivist language and terminologies through out the document.

6. Under the research paradigm and question, it is stated that “The main goal of the article is to analyze the meanings people attach to mental health based on their experiences”. More context is needed to understand what is meant by ‘their experiences’. What experiences are being referred to here? Lived experiences?

• In the last statement of the concluding paragraph in the same section, it is stated that “This research is therefore intended as a sample of the views of Somali people living in London, where the opinions and views expressed in this article are developing and changing.” The paper is exploring perceptions of mental health. So what does it mean to say opinions and views expressed are ‘developing’ and ‘changing’?

7. Under the results section, the first statement reads “This research aimed to gather better understanding of the perspectives of mental health among Somali people living in London.” What is the meaning and significance of ‘better understanding’ in this context?

8. The manuscript needs major revisions in ways that present philosophical, methodological, and interpretive clarity. In its current state, it is difficult to understand the significant contributions that this manuscript makes to current knowledge, partly because of these inconsistencies.

6. PLOS authors have the option to publish the peer review history of their article (what does this mean?). If published, this will include your full peer review and any attached files.

Reviewer #1: **Yes: **Bulbul Siddiqi

Reviewer #2: No

---

## [Author Response · Author response to Decision Letter 0]

23 Jul 2024

We have added our responses to reviewer and editor comments in the file entitled Gonzalez et al response to reviewers.

---

## [Decision Letter · Decision Letter 1]

9 Aug 2024

‘You are too much in this modern world, that’s why you are like this’: Understanding perceptions of mental health among Somali women in London.

PONE-D-24-07335R1

Dear Dr. Willott,

We’re pleased to inform you that your manuscript has been judged scientifically suitable for publication and will be formally accepted for publication once it meets all outstanding technical requirements.

Kind regards,

Lily Kpobi, Ph.D.

Academic Editor

PLOS ONE

Reviewers' comments:

Reviewer's Responses to Questions

**Comments to the Author**

1. If the authors have adequately addressed your comments raised in a previous round of review and you feel that this manuscript is now acceptable for publication, you may indicate that here to bypass the “Comments to the Author” section, enter your conflict of interest statement in the “Confidential to Editor” section, and submit your "Accept" recommendation.

Reviewer #1: All comments have been addressed

Reviewer #2: All comments have been addressed

2. Is the manuscript technically sound, and do the data support the conclusions?

Reviewer #1: Yes

Reviewer #2: Yes

3. Has the statistical analysis been performed appropriately and rigorously? 

Reviewer #1: N/A

Reviewer #2: N/A

4. Have the authors made all data underlying the findings in their manuscript fully available?

Reviewer #1: (No Response)

Reviewer #2: Yes

5. Is the manuscript presented in an intelligible fashion and written in standard English?

Reviewer #1: Yes

Reviewer #2: Yes

6. Review Comments to the Author

Reviewer #1: I have seen that the author addressed the comments of the reviewers. I suggest to accept the manuscript.

Reviewer #2: I have reviewed the corrections made in the manuscript and I am satisfied with the revisions. The manuscript is stronger now.

7. PLOS authors have the option to publish the peer review history of their article (what does this mean?). If published, this will include your full peer review and any attached files.

Reviewer #1: **Yes: **Bulbul Siddiqi

Reviewer #2: No

---

## [Editor Report · Acceptance letter]

15 Aug 2024

PONE-D-24-07335R1 

PLOS ONE

Dear Dr. Willott, 

I'm pleased to inform you that your manuscript has been deemed suitable for publication in PLOS ONE. Congratulations! Your manuscript is now being handed over to our production team.

Kind regards, 

on behalf of

Dr. Lily Kpobi 

Academic Editor

PLOS ONE